# Learning from the Past in the Transition to Open-Pollinated Varieties

**Dana Freshley [1] and Maria Mar Delgado-Serrano [2],\*** 

[1]  Department of Agricultural Economics, Ghent University, 9000 Gent, Belgium; dana.freshley@ugent.be
[2]  Department of Agricultural Economics, WEARE, ETSIAM, Universidad de Córdoba, 14071 Córdoba, Spain
\*  Correspondence: mmdelgado@uco.es

**Abstract:** In Nepal, hybrid seed introduction caused major yield gains in agricultural production, but at high environmental costs. The development of high-yielding open-pollinated varieties has spurred hope for more sustainable production systems. Nepal's government is interested in boosting their use. This research aimed to identify farmer perceptions on the factors behind the past adoption of hybrid seeds in order to propose guidelines to support the diffusion of open-pollinated varieties. Using in-depth interviews, a focus group and participant observation we explored how the process of hybrid seed diffusion has taken place in Panchkhal valley, a representative case study. Social influencers such as change agents, peers, neighbours and seed sellers, as well as economic gains emerged as major reasons for hybrid seed adoption. We learnt that the role of external agents, on which most of the governmental strategies rely, changed over time as peer-based strategies became essential after the diffusion process started. To boost the adoption of open-pollinated seeds, efforts should concentrate in developing high-yielding varieties, engaging early-adopters among influential caste members and seed sellers, distributing seeds to both disadvantaged and wealthy farmers, and using different instruments, from institutional agencies to NGOs, to deliver training on sustainable farming techniques and their economic and environmental advantages.

**Keywords:** hybrid seeds; local seeds; open-pollinated varieties; farmer perceptions; agricultural technology adoption; change agents; Nepal

## 1. Introduction

The developing world witnessed an extraordinary period of agricultural productivity growth over the past 50 years, but at the expense of significant environmental degradation and the uneven distribution of the benefits. During the Green Revolution, agriculture became the primary driver of economic growth and poverty reduction in developing countries [1]. Moreover, the numerous crop improvement initiatives at the time allowed for breeding materials and knowledge to be widely available and used [2].

The adoption of synthetic fertilisers, pesticides and high-yielding hybrid seeds spurred considerable improvements in agricultural productivity, food quality and efficiency [3]. Between 1960 and 2000, yields for all developing countries rose 208% for wheat, 109% for rice, 157% for maize, 78% for potatoes and 36% for cassava [4]. Agricultural modernisation was found to have a positive effect on measures of economic growth, human development and well-being [5]. Even the poorest countries such as Nepal were able to capture spill over benefits from neighbouring countries and from international agricultural research centres [6].

On the other hand, modern agriculture has negatively impacted the environment due to the large-scale use of chemical inputs and the deterioration of natural habitats [7–9]. Agricultural growth has been associated with the heavy use of fossil fuels, water, soil degradation and a variety of residue

problems in the surrounding environments [1]. In areas of high intensification, farmers are seeing environmental damage due to high levels of fertiliser and pesticide use [10].

Environmental consequences heightened due to lack of knowledge and misuse of hybrid seeds, fertilisers and pesticides by farmers. The inefficient use of inputs in developing countries contributed to disproportionate environmental degradation compared to developed countries [11]. In many cases, the local policy environments promoted the overuse of inputs and the expansion of cultivation into areas that could not sustain a high level of intensification, such as mountain slopes [4].

Moreover, the success of hybrid seed adoption has been uneven. Green Revolution technology which was based on the intensification of flat and arable areas has not contributed to poverty reduction in less-favourable production environments. In South Asia, the poorest areas and hilly regions that relied on rain-fed agriculture were the slowest to benefit from Green Revolution technology [4]. Over time new technologies required more mechanisation and capitalisation and higher levels of education, which has disadvantaged small farmers [12]. Larger farms have benefitted more from technological advancements, whereas yield gains halted in areas with limited access to agricultural inputs and infrastructure [13].

The urgent need to reverse this environmental degradation and social inequity is fostering the emergence of new approaches to sustainable agriculture. Improving efficiency in the use of agricultural inputs was named by the FAO as the first step to the transition to sustainable agricultural and food systems [14]. New projects and policies in local crop improvements, agroforestry and soil conservation, conservation agriculture and integrated pest management have already benefited millions of farmers [15].

Additionally, the development of high-yielding open-pollinated varieties (which are plants naturally pollinated by birds, insects or wind as opposed to hybrids which are not made to be pollinated naturally) is on the rise in developing countries and several governments are investing in the development of these seeds and promoting their use among farmers. Several examples can be identified of this new trend. In Mali, the International Crops Research Institute for the Semi-Arid Tropics (ICRISAT) partnered with the national and regional authorities to register 13 open-pollinated varieties whose use by farmers expanded under the hybrid sorghum programme [16]. In Portugal, a participatory breeding programme called the VASO project allowed for farmers to coordinate with breeders in the development of high-yielding open-pollinated varieties [16]. In India, the government is actively promoting locally bred open-pollinated varieties by capping the prices that international seed companies are able to charge [17].

Open-pollinated varieties require less inputs such as fertiliser and pesticides, and are less expensive and more affordable for low income farmers [18]. Further, reducing the reliance on hybrid seed imports and developing seeds adapted to the local environment provides opportunities for poverty reduction in disadvantaged regions.

The agricultural sector holds the largest portion in Nepal's economy. Agriculture is the main source of food, income and employment and contributes about 35% to the Gross Domestic Product (GDP). The sector is characterised by relatively low yields compared to neighbouring countries. Land is mainly allocated to grain staples (rice, maize, wheat, millet, barley and buckwheat), despite fruits and vegetables showing relatively higher yields and higher growth in consumption [19]. The competitiveness of this sector is decreasing, as the country has evolved from being a net food exporter to a net food importer [20].

To increase agricultural production and to diversify the agricultural base, the government of Nepal has boosted modernisation policies such as irrigation, the use of fertilisers and pesticides, the introduction of new technologies, new high-yielding varieties of seeds and the provision of credit. The use of hybrid seeds has increased dramatically in the past 20 years and is now common in vegetable crops, maize and rice [21,22], due to the ease and availability of hybrid seeds in rural agrovet shops and the commercialisation of seeds developed by multinational seed companies. However, this market remains unregulated, as the informal seed system accounts for nearly 90% of the total seed requirements

of farmers [22]. Given the open border with India, large quantities of unregistered hybrids are freely traded in Nepal through a huge network of agrovets [23]. The unavailability of national high-yielding open-pollinated varieties also forces farmers to use hybrid seeds in particular vegetable crops [21].

The government is interested in reversing the dependency on hybrid seed imports, aggravated by the country's low seed breeding capacity. To further formalise the Nepal seed sector and to reduce the need for seed imports, the government aims to develop locally-adapted hybrid and open-pollinated varieties, and to make these seeds available when and where needed. Indeed, the country aims to release 423 new open-pollinated varieties and 60 new hybrid varieties by 2025 [22].

The current flow of open-pollinated varieties to local producers is markedly unexplored and un-regulated in Nepal [24]. A recent analysis on the adoption of open-pollinated varieties in Nepal found that 83.3% of farmers obtained their open-pollinated varieties from saving their own seeds or from fellow farmers, whereas only 16.7% of them obtained these seeds from a government organisation like the local District Agriculture Office [25]. Among the reasons explaining the low number of farmers having access to these seeds might be that most of these seeds are distributed by the Nepalese Department of Agriculture, which is mandated to deliver extension services related to crops and fisheries, but only delivers extension services to farmers who are members of farmer groups or cooperatives [26].

In this scenario, the objective of this research is to identify farmer perceptions on the factors behind the successful adoption and spread of hybrid seeds in the country, in order to analyse their utility to support the diffusion of these new open-pollinated varieties and promote a shift towards more sustainable production models. By identifying the biggest technological and social influencers on the adoption of hybrid and local seeds, we might offer relevant insights to policymakers on the most effective methods and actions for the adoption of sustainable agricultural technologies by farmers in the country.

Analysing farmers' perceptions and behaviour and the factors that influence different types of farmers can offer lessons for the adoption of open-pollinated varieties in the future. Understanding the strongest methods of influence on current seed use offer insights on what the highest potential might be to promote a change in seed use, especially towards more sustainable practices and open-pollinated varieties over the next decades.

On a theoretical level, we used the technology diffusion approach, a widely-used method to analyse how new agricultural technologies spread and are adopted by farmers [27]. This approach adheres to adoption as a social process that depends on the specific relationships, social network structures, local organisations and farmers' perceptions of innovation [7,28,29]. However, as modern agricultural technologies make their way into developing countries, social and cultural factors often act as a barrier to change.

Following the large body of studies showing that social learning is an important element to innovation adoption and diffusion [29–34], this study analyses the social processes that promote or inhibit the adoption of local and hybrid seeds by farmers, paying close attention to the surrounding organisations and institutions influencing farmers. We also aimed to unveil other potential factors that influence technological adoption, such as economic and market factors, product characteristics and environmental factors. We consider both hybrid and local seeds as "new" agricultural technologies, because the local seeds being promoted in Nepal are mostly improved, high-yielding open-pollinated varieties.

## 2. Methods

We used the case study, which is a common methodology in social sciences. This method involves a close, in-depth and personal examination of a particular case and its contextual surroundings [35] and has a very wide scope, allowing for an in-depth understanding and investigation of phenomena involving human affairs within their real-life context—especially when they are context-specific, as technology adoption has been found to be [36–38].

Case studies use an empirical inquiry followed by a descriptive and exploratory analysis. Our aim was to gather perspectives on hybrid and local seed use and analyse the opinions of farmers and key informants, avoiding subjective bias, which is a common mistake in case study research [39]. We used an exploratory method to analyse the farmers' perceptions of their technology adoption, following several studies that have found that farmers' perception on modern technologies significantly influences adoption decisions [40–42]. A farmer's perception may be determined by experience, extension visits, knowledge and other conditions [40], and farmers' experience with one type of technology will likely affect their perception of future technologies and eventually the decision to adopt [43]. Adhering to the technology diffusion approach, we also explored the social and cultural factors that might act as barriers to change.

*2.1. Study Site*

We selected Panchkhal valley, 40 km from Kathmandu, as our case study site, since it was the first region in the country to adopt hybrid seeds and industrial farming practices around 20 years ago, and is now one of the most farming-intensive regions of Nepal [44] (confirmed by the personnel communication and the Executive Director of the Center for Environmental and Agricultural Policy Research, Extension and Development, CEAPRED). The seed replacement rate (SRR), which is a measure of how much of the total cropped area is sown with certified seeds in comparison to farm saved seeds, is estimated to be around 60% in Panchkhal valley, but only 5% in the surrounding remote mountain regions, which confirmed the reliance on hybrid seeds in this area [45]. Moreover, other factors such as market accessibility to Kathmandu and favourable access to the seed and fertiliser subsidy by Panchkhal valley farmers [45] facilitated the use of hybrid seeds.

The region supplies vegetables to the capital city, which has opened market opportunities and spurred economic growth. Maize was the most prominent crop grown with hybrid seeds, with rice, beans, tomatoes and cauliflower as other major hybrid crops. Even though the area was facing a high agricultural transition, both wealthier and more disadvantaged farmers could be found there, which increased the interest of the case study.

The municipality of Panchkhal has around 40,000 inhabitants, and the Kavrepalanchok District, in which it lies, has around 80,651 inhabitants [46,47]. Kavrepalanchok has an average household size of 4.73 people and a literacy rate of 69.8%. The main religion is Hindu, to which 63% belong. Other religions include Buddhism (35%) and Christianity (2%) [47]. Castes, which are a form of social hierarchy within the Hindu religion, play an important role in the life of Nepalese people.

*2.2. Data Collection and Analysis*

Data were collected using 29 interviews with farmers, 5 interviews with key informants, 1 focus group and participant observation. We conducted semi-structured interviews with farmers who had adopted hybrid seeds and were farming for commercial purposes. Interview questions on the factors influencing seed use were open, aiming at gathering farmers' perceptions on different influencers—from social to economic, environmental and institutional factors.

General questions about local and hybrid seeds were asked to gain a better understanding of their use of both. These included questions about the differences in farming methods, vegetable varieties, pesticide use and the challenges of local and hybrid seed use.

The interviews were conducted in two separate locations. The first was used for farmers residing in the low-lands of the valley, who had abundant flat arable land and adequate road connection to Kathmandu. Second, farmers in the surrounding hillsides were interviewed to compare the responses of the farmers in the valley with those of farmers residing in more remote villages, with smaller farms and poor access to city facilities. We interviewed 17 farmers in the valley and 12 farmers in the hillsides. The farmers were purposefully identified by a key member of the community and subsequently via snowball sampling [48]. A gender- and age-sensitive approach was applied in the selection of respondents to balance men and women and different range ages, when possible.

Key informants such as NGO and government workers and seed sellers were identified from the interviews with farmers and interviewed to gain a broader understanding of the influencing process of hybrid seed use. Five key informants were interviewed: the Deputy Major of Panchkhal, a worker at the local Crop Development Centre, the Executive Director of CEAPRED, a local grocery shop keeper who sold seeds and an agrovet shop keeper whose specialty was selling seeds, fertiliser, pesticides, animal feed and veterinary supplies.

To gather information on why farmers chose sustainable farming methods, a focus group took place in the hillsides, in a "climate smart village" where farmers were participating in an experimental farming approach that used natural pesticides and organic techniques, in partnership with CEAPRED. This offered a unique study site where farmers were practicing alternative farming methods, which contrasted with the pesticide-heavy practices in much of Panchkhal.

A qualitative analysis was conducted using deductive coding in the software programme NVivo, version 10 [49]. Codes were created for every influencer of local and hybrid seed use by one researcher, identifying the different factors mentioned by each respondent and including them in Tables 1 and 2 each time a respondent mentioned them. Then, the major influencers on local and hybrid seed use that emerged were analysed and compared between both the valley and hillside sample groups. Influencers were grouped into four categories: external influencers, internal influencers, economic factors and environmental factors. Due to the important role they played in our analysis, we separated external social influencers (we called them *change agents*, identified in the literature [27] as those belonging to external institutions or organisations) from internal social influencers (neighbours, cooperatives, local seed sellers and social caste networks).

Among the market and economic influencers of seed use, categories were identified such as productivity, market facilities and crop appearance. Finally, a major topic mentioned by the interviewees were the environmental problems associated with seed use, such as soil degradation, misuse of pesticides and loss of taste and nutrition.

All these different categories were extracted from the survey answers, analysed and compared between the local and hybrid seed use, as well as between valley and hillside respondents. Key informant interviews were analysed separately. Living with a hybrid seed farmer and his family in one of the villages for a consecutive period during field research allowed for further knowledge to be gathered through close participant observation.

## 3. Results

From our interviewed sample, the farmers' ages ranged from 33 to 65 years old, with an average age of 50 years old. A total of 24 male and 5 female farmers were interviewed. Most farmers owned farmland (69%), while the rest rented at least a portion of farmland (31%). The average farm size for valley inhabitants was 1.6 hectares, and the average farm size for hillside inhabitants was 0.9 hectares, both higher than the country average (0.7 hectares) [50]. Valley inhabitants belonged mainly to the highest social caste, Brahmin, whereas hillside villagers belonged mainly to a middle social caste, Janajati.

Using the concept of diffusion process, we analysed the main influencers on the farmers' adoption of hybrid seeds, and also what influenced their decision to use hybrid seeds or to continue using local seeds, according to their perspectives. Tables 1 and 2 summarise our findings for the valley and hillside farmers, respectively. Thereafter, we use the common narrative style of case studies [39] to describe the richness of the answers found, which are not reducible to quantitative comparisons. The types of crops were similar for both hybrid and local cultivation, with crops like maize, beans, tomatoes and cauliflower using both types of seeds.

**Table 1.** Main influencers in the use of hybrid and local seeds among valley respondents.

| | | Land Used (Percentage) | Change Agent Influencers * | Internal Influencers ** | Economic Factors *** | Environmental Factors **** |
|---|---|---|---|---|---|---|
| Farmer 1 | Hybrid | 67% | A | E, F | I, J | N |
| | Local | 33% | | | | |
| Farmer 2 | Hybrid | 88% | | | I | |
| | Local | 12% | | | | |
| Farmer 3 | Hybrid | 100% | A | E, H | J | M |
| | Local | 0% | | | | |
| Farmer 4 | Hybrid | 100% | | E, H | I, J | M |
| | Local | 0% | | | | |
| Farmer 5 | Hybrid | 86% | A | G, H, J | | M |
| | Local | 14% | | | | |
| Farmer 6 | Hybrid | 83% | | E | J | |
| | Local | 17% | | | | |
| Farmer 7 | Hybrid | 88% | | | I, J | L, M |
| | Local | 12% | | | | |
| Farmer 8 | Hybrid | 80% | | E | I, J, K | |
| | Local | 20% | | | | |
| Farmer 9 | Hybrid | 100% | | E, H | I, J | N |
| | Local | 0% | | | | |
| Farmer 10 | Hybrid | 67% | | E. H | I, J | M, N |
| | Local | 33% | | | | |
| Farmer 11 | Hybrid | 73% | B | G, H | I, J | |
| | Local | 27% | | | | |
| Farmer 12 | Hybrid | 70% | | E, G, H | I, J | |
| | Local | 30% | | | | |
| Farmer 13 | Hybrid | 75% | | F, H | J, K | |
| | Local | 25% | | | | |
| Farmer 14 | Hybrid | 67% | | F | I, J, K | |
| | Local | 33% | | | | |
| Farmer 15 | Hybrid | 71% | | H | I, J, K | |
| | Local | 29% | | | | |
| Farmer 16 | Hybrid | 80% | B | E, H | I, J | M |
| | Local | 20% | C | | | |
| Farmer 17 | Hybrid | 14% | | E, G | I, J | |
| | Local | 86% | | | | |

* Change agent influencers: A: District Agricultural Office; B: CEAPRED; C: Crop Development Centre; D: Indian charity organisation. ** Internal influencers: E: Neighbours; F: Cooperatives; G: Seed sellers; H: Caste networks. *** Economic factors: I: Productivity; J: Market facilities; K: Crop appearance; **** Environmental factors: L: Soil degradation; M: Misuse of pesticides; N: Loss of taste and/or nutrition.

**Table 2.** Main influencers in the use of hybrid and local seeds among hillside respondents.

| | | Land Used (Percentage) | Change Agent Influencers * | Internal Influencers ** | Economic Factors *** | Environmental Factors **** |
|---|---|---|---|---|---|---|
| Farmer 18 | Hybrid | 43% | | E, H | I | |
| | Local | 57% | | | | |
| Farmer 19 | Hybrid | 75% | | H | I | |
| | Local | 25% | C | | | |
| Farmer 20 | Hybrid | 58% | B | H | | M |
| | Local | 42% | C | | | |
| Farmer 21 | Hybrid | 75% | | E, G, H | I | |
| | Local | 25% | | | | |
| Farmer 22 | Hybrid | 57% | | E | I | L |
| | Local | 43% | | | | |
| Farmer 23 | Hybrid | 100% | | E, G | I | |
| | Local | 0% | | | | |
| Farmer 24 | Hybrid | 67% | D | G | | M |
| | Local | 33% | | | | |
| Farmer 25 | Hybrid | 40% | | | | N |
| | Local | 60% | C | | | |
| Farmer 26 | Hybrid | 14% | | E | | M |
| | Local | 86% | | | | |
| Farmer 27 | Hybrid | 14% | | | I | |
| | Local | 86% | | | | |
| Farmer 28 | Hybrid | 50% | | | | M |
| | Local | 50% | | | | |
| Farmer 29 | Hybrid | 67% | B | G, H | | L, M |
| | Local | 33% | | | | |

* Change agent influencers: A: District Agricultural Office; B: CEAPRED; C: Crop Development Centre; D: Indian charity organisation. ** Internal influencers: E: Neighbours; F: Cooperatives; G: Seed sellers; H: Caste networks. *** Economic factors: I: Productivity; J: Market facilities; K: Crop appearance. **** Environmental factors: L: Soil degradation; M: Misuse of pesticides; N: Loss of taste and/or nutrition.

### 3.1. External Influencers

Four local institutions operating in Panchkhal valley that were trying to influence farmers to use local or hybrid seeds, or change agents, were named as influencers by the farmers: the District Agricultural Office (DAO), CEAPRED, an Indian charity organisation and the Crop Development Centre (CDC). The DAO, CEAPRED and the Indian charity organisation were hybrid seed promoters and the CDC was a local seed promoter. The DAO and CDC were governmental organisations and the other two were NGOs.

These change agents showed a moderate capacity of influence (41%), being more prominent among hillside respondents. Out of the valley respondents, 29% said they were influenced by at least one change agent, and out of the hillside respondents, 42% said they were influenced by at least one change agent.

The methods that change agents used to influence farmers were different, each with their own tactics to fit their motives. The DAO, part of a network of nation-wide governmental farmer extension services, was instrumental in hybrid seed distribution and training about 20 years ago, when hybrid technology was beginning to arrive in Nepal. Farmers immediately saw the production increase in their fields. Farmers who received training from the DAO said the training was aimed at teaching farmers how to systematically plant seeds in a line and a certain distance apart. Farmers said they still plant using these methods because of this training. The DAO was mentioned as an influencer by farmers in the valley but not by farmers in the hillsides.

CEAPRED, an agricultural NGO based in Kathmandu, was also involved with early hybrid seed adoption in Panchkhal valley according to farmers. The Executive Director of CEAPRED said that around 25 years ago, the NGO introduced hybrid maize in the valley, but they no longer implement programmes in Panchkhal valley. The organisation is now more focused on promoting sustainable agriculture in other areas of Nepal. Farmers only spoke of the past involvement of CEAPRED and not of current involvement, further explaining the NGO's shift away from Panchkhal valley and to other areas. Some farmers in both the valley and the hillsides mentioned CEAPRED as an influencer, indicating that CEAPRED's involvement was more widespread than that of the DAO.

The CDC, an open-pollinated variety breeding centre located in Panchkhal valley, was the only change agent mentioned by farmers that did not promote hybrid seeds. According to the CDC's key informant, the main purpose of the centre is to preserve local seeds. The centre distributes high-yielding local seeds that are bred in Nepal and adapted to the local environment, such as garlic, ginger, fenugreek, rice and potatoes. Trainings are not focused on industrial farming methods but rather on how to produce sustainably from high-yielding, open-pollinated local varieties, i.e., worm composting, organic manure preparation and integrated pest management techniques.

The CDC targets rural and hilly regions where farmers are poorer and cultivate mostly local crops, mainly because of the challenge of persuading farmers in Panchkhal valley to cultivate local seeds. The CDC's key informant said that farmers in the valley prefer hybrid varieties and do not have the patience to wait for local varieties to become ripe. He explained that farmers have a limited knowledge of open-pollinated varieties and lack information on their potential revenues in the long term due to their lower costs. The CDC was named only once by a valley respondent, indicating that the centre is indeed focused in hilly regions.

The Indian charity organisation Nabajunti Kindra was only named as an influencer in the hillsides. The organisation distributed hybrid seeds at a local school after the April 2015 Nepal earthquake, which considerably helped farmers who could not otherwise afford to buy hybrid seeds.

### 3.2. Internal Influences

The spread of hybrid technology was much more accelerated by the influence of peers in both the valley and hillsides. When asked what their major influences were to adopt hybrid seeds, 52% of all respondents said their neighbours played some part in their decision to adopt. Among valley respondents, 59% said they were influenced by neighbours or members of their own social caste

groups, and among hillside respondents 42% said the same, suggesting that these influences were slightly stronger among high hybrid seed adopters.

Most farmers said they were influenced by their neighbours by seeing the productivity increase first-hand, or the cob size, or healthy-looking crops in their neighbours' fields. Some farmers expressed jealousy when seeing bigger and taller crops in neighbours' fields. One farmer in the valley said the entire village switched to hybrid so it no longer made sense for him to continue using local seeds. Another farmer said there was a cultural shift towards agriculture expansion taking place in Panchkhal and she switched to hybrid seeds because of this local culture of adding more commercial crops.

As Nepal's culture and Hindu caste system are deeply rooted in tightly organised kin-like communities, this social organisation played an important role in influencing seed use. Farmers in the valley who cultivated mainly hybrid crops belonged to higher and wealthier castes and were more confident in their answers and proud of their farms. This may have positively influenced their access to, and use of, hybrid seeds. Farmers in the hillsides, belonging to lower castes, did not have the wide network that higher castes enjoyed. Hillside respondents also had less economic capacity to afford hybrid seeds, but were still strongly influenced by their own caste networks.

Another important social influence on the farmers' seed use was the impact of local seed sellers. Seed sellers in Panchkhal ranged from grocery shops to agrovets and to independent "middle men" who came from Kathmandu. Farmers and seed sellers had very close relationships; in many cases farmers, would choose to buy from certain seed sellers because of personal rather than commercial reasons. Around one third of farmers—24% in the valley and 33% in the hillsides—indicated that they were either neighbours, family relatives or had years of friendship with their seed sellers. Seed seller influence was slightly higher among hillside respondents, but both groups spoke of a strong trust system between themselves and their seed sellers. Several farmers said their seed sellers gave seeds in credit, allowing farmers to receive seeds at the beginning of the season and pay back sellers after harvest. Some farmers also mentioned having a business relationship with seed sellers in which farmers made buying and selling decisions based on prices alone. This was slightly more common among valley respondents, but less common overall.

Interviews with two seed sellers provided further insight into the diffusion of hybrid varieties and the relationships between farmers and seed sellers. Both seed sellers brought seeds from large wholesale shops in urban centres that imported seeds from India, China and Thailand. They both said that they choose which seeds to sell mostly based on what farmers in Panchkhal demanded, but sometimes they would bring new seed varieties back to the village for farmers to experiment with. Both mentioned very close and personal relationships with farmers and said that most farmers operated within a tight social and economic network and went to the same seed seller for every purchase.

Finally, cooperatives also encouraged farmers in the valley to adopt hybrid seeds, by providing small-scale farmers with access to affordable hybrid seeds. According to the Executive Director of CEAPRED, cooperatives have emerged in Panchkhal valley as a way to mobilise and empower poor farmers and expand their market potential. Cooperatives are formed informally by groups of small-scale farmers and registered by the government after they demonstrate enough market potential. The local DAO supplies hybrid seeds to formalised cooperatives and then monitors farmers to make sure the seeds are properly utilised.

Cooperatives have helped many small-scale farmers, but it is mostly the government that determines their level of success. Cooperatives located closer to urban areas have more money and are more influenced by the government. Farmers who belonged to cooperatives said they received seeds regularly, but not enough. Hence, they had to rely on other sources to get the remaining seed they needed. One farmer said she had been receiving seeds from a cooperative for the past 16 years, but the seeds had been less and less productive, so that year she chose to purchase from an agrovet instead.

*3.3. Economic and Market Factors*

Almost all interviewed farmers in both the valley and the hillsides used both hybrid and local seeds. Local crops were used for subsistence production and hybrid crops were used for commercial production. Farmers in the valley cultivated a higher percentage of hybrid crops and sold a higher amount of crops compared to farmers in the hillsides. These farmers had larger plots of land and more mechanised farming methods.

Yield had a major relevance in the decision to use hybrid seeds. Most respondents said their production increased drastically with their switch to hybrid seeds (66% on average, but 76% for valley farmers). Hybrid seed use was directly linked to economic growth, as farmers who were once unable to feed their families were now able to do it and have a surplus to sell. Many farmers explained a two-fold yield increase in crop production on the same plot of land after switching to hybrid. Some farmers in the valley also mentioned the crop appearance and homogeneity of crops (colour and cob size) as important factors to switch to hybrid seeds.

Market networks and facilities to buy hybrid seeds, along with the consumer demand for commercial vegetables, have been paramount to valley farmers, while none of the hill farmers mention these factors. Panchkhal valley was much more connected to Kathmandu, both economically and socially, than the surrounding hillsides. The high demand for vegetables created a dynamic business environment in the valley, where trucks would arrive from Kathmandu every morning with "middle men" to buy fresh produce and sell seeds. Many valley respondents said that local grocery store owners drove directly to their farms to collect products. Very few farmers in the valley had to deliver grains or products to grocery stores themselves.

Farmers in the hillsides travelled much further to buy seeds and sell products. Although there were a few agrovets and grocery stores located in the hillsides, many farmers chose to travel further to urban centres to buy seeds. Hillside respondents delivered products to buyers themselves more often than buyers coming to their farms to pick up products. Several farmers in the hillsides delivered their grains and products to wholesale shops or to "middle men" in urban centres.

*3.4. Environmental Factors*

The overuse of chemical pesticides and fertilisers was by far the largest environmental concern among both valley and hillside respondents. It was a point of concern for 35% of valley respondents and 42% of hillside respondents. The biggest difference in the farming techniques between local and hybrid seed use was the spreading of chemical pesticides and fertilisers. Farmers said that the planting technique had not changed, but with hybrid crops they were compelled to use chemical inputs. A big challenge for farmers was their lack of knowledge to naturally retain nutrients in their soil, and even though they preferred not to use chemical nutrients, it was easier for them.

There was a growing awareness among farmers about the negative effects of pesticides on their health. In response to several farmers experiencing health issues associated with pesticide use, the government issued a ban on liquid pesticides and encouraged farmers to wear gloves when applying the product. Despite this, several farmers said they were still unsure about how to effectively use pesticides. In order to stay safe, wealthier farmers would pay lower caste members to spay pesticides in their fields.

Soil degradation was another environmental challenge mentioned explicitly by farmers, 6% in the valley and 17% in the hillsides. They said that the productivity and quality of their soil had gone down due to the intense use of chemical fertilisers and pesticides. The contrary was true for farmers residing in the "climate smart village", who were learning how to use natural forms of fertilisers and pesticides. These environmental training programmes for farmers, organised by CEAPRED and The International Centre for Integrated Mountain Development (ICIMOD), encouraged farmers to use natural pesticides such as cattle urine and integrated pest management (IMP) in order to improve soil quality.

Another personal challenge for farmers was the loss of taste and nutrition of hybrid seeds. Farmers mentioned that hybrid products significantly lacked the taste and nutritional quality of local varieties.

For this reason, even after adopting hybrid seeds for commercial cultivation, many farmers kept a small plot of land to grow local crops for their own consumption. Farmers tended to use hybrid crops to feed their livestock but not to feed themselves.

## 4. Discussion

In this section we highlight the most feasible possibilities along with the potential barriers for open-pollinated seed dissemination in Nepal, based on our analysis of Panchkhal valley.

### 4.1. The Role of Social Networks in Influencing Seed Adoption

Change agents were the initial influencers in the adoption of hybrid seeds. They acted as mediators and facilitators linking communities to new trends [51]. Connectedness and relationships between farmers and change agents has been found to influence their decisions to innovate [27]. Training and seed distribution by change agents played an important role in hybrid seed adoption. Trainings were one of the most prominent ways whereby farmers learned about hybrid seeds when they first arrived in Panchkhal valley. Many farmers still used the methods they learned during these trainings, which shows that local seed trainings can have a long-lasting impact.

If improved local seeds are going to be promoted, training on the agricultural practices, seed distribution and information on the advantages of local seeds, together with a promotion of their environmental advantages, are necessary. Widespread regional trials and trainings for open-pollinated varieties managed by CIMMYT are taking place in eastern and southern Africa [52], but large-scale trials and trainings for open-pollinated varieties are limited in Asia. In Nepal, the CDC is already leading formal trainings on open-pollinated varieties, but these are targeted at farmers in poorer, hilly regions.

In this research we found that change agent influence in the valley was much more prominent when hybrid seeds were first being introduced to the area. But this trend has now reversed, as results show that after the innovations spread, change agent influence weakened and peer influence became much higher. After the initial stages of adoption, farmers are more likely to gather information horizontally from people similar to them, and new ideas are more easily adopted when they come from members of similar social groups [29,53].

Results show that farmers in the valley have shifted from organisational support to relying on other sources for issues regarding hybrid seeds, and that change agents in Panchkhal have moved into poorer, hilly regions, where we identified change agents promoting both hybrid and local seeds. This evidence is congruent with what other researchers such as Carey [54] and Rogers [27] have illustrated; that farmers are more connected with organisations during the early stages of adoption and rely on their personal experiences and peers during the late stages in adoption.

Seeing that most of the current diffusion of open-pollinated varieties is organised by change agents, this outcome highlights the importance of rethinking this strategy. The strong social ties that exist in Panchkhal offer the potential for dissemination. The close-knit communities are advantageous for the spread of new technologies. The majority of farmers interviewed grew up in the same village where they currently live and rarely travelled outside the village. Farmers belonging to the same caste group often lived next to or near each other. In these communities, it is likely that once new high-yielding local seeds are introduced, the news of new technologies will spread quickly through sharing personal experiences with seed sellers, neighbours and community members.

Our findings are consistent with those of other authors [55,56], who found that membership of kinship and friendship networks positively influenced the adoption of open-pollinated varieties and conservation agriculture. Social networks are especially important for the diffusion of new technologies in the absence of formal markets. In Tanzania, social networks played a significant role in the spread of information about open-pollinated varieties but not maize hybrids, which were sold by private seed companies [55].

In Nepal, higher castes have a higher level of power and influence and larger social networks [51]. The extensive networks of farmers belonging to higher castes and their connections to large input distributors probably played a role in their ability to use the latest agricultural technologies. For local seed adoption to succeed, the involvement of wealthy, high caste farmers, such as those in the valley, and the involvement of seed sellers will be essential. For this strategy to be effective, early adopters and innovators recognised as opinion leaders need to be identified and engaged to test these open-pollinated varieties and to influence farmers to switch to these seeds. They have the highest potential to influence other farmers by using informal information channels [33].

*4.2. Main Economic and Market Factors Influencing Seed Adoption*

The key factors influencing the adoption of hybrid seeds were productivity and higher economic turnover, together with standability and the uniform look of the fields. Traditional local seeds are adapted to the local environment and are more sustainable, but do not match the exceptional high yields of hybrids.

For this reason, efforts to develop and promote high-yielding local seeds are expanding in several developing countries. In Kenya, there are a number of participatory local crop improvement projects involving government demonstration and experimentation fields [57]. India's government is actively promoting new homegrown cotton varieties, and thousands of cotton farmers have switched to new local varieties [17]. Open-pollinated varieties are also being used in Italy to preserve the globe artichoke, which is threatened by hybrids [58].

High-yielding local varieties have shown success especially in small-scale farming systems with small profit margins, where farmers can save money by reducing the usage of chemical inputs. The seed price and the fertiliser and pesticide input requirements for hybrid cultivation are so costly that many farmers need access to credit to make these pre-season investments [59]. Local seed cultivation has drastically lower input requirements than hybrid cultivation. Studies have shown that with little or no input of synthetic fertilisers, the yield of open-pollinated varieties can be compared to that of hybrids [60,61] and that the second-generation seeds of these open-pollinated varieties can result in the same yield as the first generation, unlike hybrids [18]. In South Africa, farmers preferred open-pollinated varieties to hybrids because they were less costly and required less inputs to grow [62]. These varieties are often the best choice for small scale-farmers who cannot afford the cost of hybrids and cannot pay for inputs such as fertilisers and pesticides [61]. Therefore, these seeds remain especially important in farming systems where hybrid seed and fertiliser prices are high.

Since the cost-effectiveness of open-pollinated varieties is mostly demonstrated in small-operation farms, large farms may be more averse to using open-pollinated varieties. In this research, wealthy farmers with larger farms grew more hybrid than local crops. Therefore, replacing hybrid with open-pollinated varieties in wealthy farming areas may be challenging. It may be easier for small-scale farmers, whose farming methods already somewhat match those of local seed cultivation, to adopt new open-pollinated varieties.

However, market facilities for small-scale farmers must also be improved. In this research, farmers in the hillsides faced hurdles for market accessibility such as road distance and transportation costs. Market accessibility can be enormously enhanced by improved transportation facilities and road infrastructure [42]. Additionally, developing farmer cooperatives and farmer organisations, business associations and scientific organisations that explicitly support the needs of small-scale producers is recommended by the FAO to improve smallholder access to markets [63].

The formalised system for quality monitoring and regulation of open-pollinated varieties, as well as market access to these seeds, should be improved. Currently, formal markets for open-pollinated varieties are limited, and for farmers to obtain open-pollinated varieties, they must rely on government extension services or NGOs.

Overall, the greatest potential for local variety adoption lies in the lower price, the lower need for inputs such as fertilisers, water or pesticides, the possibilities of storage and replanting in subsequent

years, the natural pest resistance and the adaptation to local conditions. Therefore, the highest return on investment may result from the use of improved open-pollinated varieties, but most farmers have limited or no knowledge of this. These advantages need to be broadly disseminated to both large and small-scale farmers and the efforts to develop high-yielding local seeds intensified, in order to boost their adoption.

### 4.3. Environmental Factors Influencing the Use of Seeds

Several environmental aspects can be mentioned in favour of the use of open-pollinated seeds. They have the potential for greater resistance to disease and insects, higher efficiency in the use of the available water and a better nutritional content [64]. Crop varieties that are adapted to their local environments are naturally resistant to damage by diseases, insects and other pests.

These characteristics are being further strengthened by plant breeders in several parts of Africa and Asia [65–67]. The Indian government has started promoting local cotton varieties among cotton farmers because they promise good yields, are pest-resistant and are much less expensive than hybrids or GMOs [68]. An increased crop yield is the primary aim of these plant-breeding programmes, but other advantages of the new open varieties that have been developed include the resistance to abiotic stressors such as drought and heat stress [65,66,69]. In this way, low-input agriculture is now beginning to come full circle: local seeds that were traditionally used for centuries before hybrids were introduced are now being improved and promoted as new sustainable technologies.

Reduced input use and pollution are other advantages of these open-pollinated seeds, because the intensive use of inputs such as pesticides and chemical fertilisers have contributed widely to the agricultural pollution of the land and surrounding water bodies [70]. The excessive use of pesticides may also lead to the destruction of biodiversity [71] and can contribute to soil contamination [72]. In the Himalayan region of India, where soil quality has decreased due to the rampant use of fertilisers and pesticides, farmers suggested that crops requiring less inputs should be given priority [73].

Another emerging concern is the toxicity of pesticide residues in food [74] and the adverse effects on the health of workers using these products [44]. As consumers in Nepal and India increasingly demand healthier eating options, farmers in Panchkhal have the potential to be supported in their transition towards more sustainable production methods. In India, the health-conscious consumer segment is growing by approximately 10–15% annually [75]. In Kathmandu, a study showed that 58% of consumers are willing to pay a 6–20% price premium for organic products [76], and another study showed that most consumers in Kathmandu were willing to buy organic tomatoes, provided they are inexpensive and certified [77]. These results open interesting avenues for both valley and hill farmers to adopt more sustainable production practices based on local seeds.

In order to boost open-pollinated seed adoption, a broader dissemination of its environmental and health benefits is needed. Additionally, trainings for farmers on proper fertiliser and pesticide use would further reduce the damage to the environmental and human health and are crucial for a successful transition to sustainable production methods. It is expected that persistent education on the safe use of pesticides will lead to a change in the attitude of farmers, and hence minimise the amount of chemicals in the environment [78,79].

### 4.4. Policies and Political Decisions to Foster the Adoption of Open-Pollinated Varieties

The globalisation of agriculture means that it will likely remain a big business, in which high yields are expected to fulfil the global demand. The local structure of the sector in developing countries can have an influence on the impact of new mechanised technologies and agricultural input use. The institutional arrangements can facilitate growth linkages for small farmers and reduce the environmental impact of improper farming methods [1].

Policy has the potential to correct the misuse of agricultural technology and the negative effects of the Green Revolution. Models for open-pollinated seed dissemination in other countries have mostly been incentivised by the government, emphasising the need for government intervention. Different

approaches are possible, from investing in the development of high-yielding local seeds, promoting and distributing local seeds or the training and supporting their use to tightening environmental policies and controlling the negative effects of hybrid cropping. In several cases, where policy incentives supported sustainable farming, farmers quickly changed behaviour and adopted these practices. For example, the removal of pesticide subsidies in Indonesia led to a dramatic rise in integrated pest management and a drop in agrochemical use [80].

Innovation design and communication pathways drive technology adoption [33]. Our research found that market networks have a very important role in seeds use, but that the government determines both the seeds distributed and the regions of dissemination. Therefore, the use of government instruments to distribute improved local seeds and expand dissemination regions might boost the acceptance among farmers. In Panchkhal, the CDC offers a great starting point for steering governmental action towards more sustainable input and seed use.

Nepalese cooperatives were also instrumental in hybrid seed adoption among poorer and small-scale farmers, suggesting that farmers belonging to cooperatives might have a higher chance of producing successfully from local varieties due to the higher level of support and expertise they receive. In Nepal, cooperatives can exist as their own entity or as associations registered by the government, the latter receiving institutional and financial support. Members of cooperatives regularly receive seeds from the government and are monitored to ensure their correct use. Thus, the government has a unique opportunity to distribute high-yield open-pollinated seeds and spread their use.

Small-scale farmers without enough market potential can benefit hugely from the access to capital and agricultural inputs that cooperatives provide. Cooperatives and group-based extension approaches are an especially effective strategy for women and small-scale farmers in Nepal [81]. However, these approaches will be challenging among wealthier farmers who own large farms, especially if open-pollinated varieties produce a lower yield than their hybrid counterparts. Governmental influence for these farmers might be better linked to the long-term benefits of open-pollinated varieties and to the awareness of the environmental problems of hybrid seeds. Some authors point out that the excessive dependence on agrochemicals in Panchkhal has led to increased vulnerability and environmental deterioration [82].

Another aspect to consider is the reliability of government policies. Many farmers in the valley were dissatisfied with the seed distribution programmes of the DAO and cooperatives and chose to purchase from agrovets instead. Open-pollinated seed distribution needs to be trustworthy and reliable, and the seeds distributed (or even sold) need to meet the farmers' expectations.

The structure of the agricultural sector and globalisation is moving towards a preference for sustainable agricultural practices. The development of markets for traditional products with better organoleptic characteristics and nutritional values can support the switch from hybrid crops for both large and small-scale farmers. Therefore, an additional avenue for the government to promote open-pollinated varieties is to develop market facilities for these products and distribute them among traditional seed sellers.

Furthermore, encouraging sustainable practices might enhance the international position of the country. Climate smart agriculture, first launched by the FAO in 2010, has met with great success, as 32 developing countries now refer to it in their Intended Nationally Determined Contributions (INDCs) [83]. Although there are a few projects on climate smart agriculture in Nepal, the country is not part of this list and could therefore further develop its plan for climate change mitigation through agriculture.

Finally, the promotion of sustainable local seed varieties is often led by small, poorly funded organisations that must compete with the power of hybrid seed promoters. To promote open-pollinated varieties, local organisations must receive institutional support and gain traction in wealthy areas.

*4.5. Main Limitations of this Research*

This research conducted a thorough and in-depth analysis of a case study. We are fully aware that these results do not represent the whole country, since we focused on a single case study and used a small sample. However, as Ragin [84] points out, the generalisability of case studies can be increased by the selection of strategic cases. We believe our sample is an accurate representation of Panchkhal valley, and that the region is considered the most popular pocket for growing vegetables using hybrid seeds in Nepal [44]. Furthermore, most of our results are backed by those found by other researchers in different contexts.

We are firmly convinced that these results might be a sound initial contribution to the task of encouraging the transition towards more sustainable farming methods, but do not expect them to be assumed as valid for Nepal as a country. A broader analysis of the situation in other areas of the country would increase the understanding of the diffusion of hybrid seeds and would allow for improved guidelines delivered to the government to shift to open-pollinated varieties.

Additionally, we acknowledge that there may be some limitations to the diffusion of innovations theory, and that social norms and the disapproval of new technologies may not outweigh the long-term benefits that a new technology brings to a society.

Finally, once the use of these open-pollinated varieties is more spread and data are available, further research will be necessary, and some techniques such as cost-benefit analysis will contribute to determine the comparative advantages of both types of seeds.

## 5. Conclusions

Learning from the past and disseminating the best practices in the diffusion of hybrid seeds in Nepal can be an important approach to promote the adoption of improved local seeds and to address the socioeconomic and environmental problems associated with hybrid cropping. The country has the opportunity to further develop its agricultural sector by investing in open-pollinated seed breeding programmes and by formally registering them to be made available to farmers. Our research identified aspects related to social, economic, environmental and institutional factors that might support this shift in Nepal.

Increasing the yields of open-pollinated varieties is essential in order to provide fair livelihoods to farmers, but it is also important to consider that economic benefits might be increased if the costs of inputs, such as seed and chemicals, are lower. A key challenge, in this regard, is to engage wealthier farmers in the use of these seeds. It might be helpful to identify innovators and early adopters among farmers and seed sellers with high influential capacity among the members of their own caste, and to involve them in trials, trainings and communication activities that display the advantages of these seeds.

The actual dissemination practices based on change agents (governmental organisations or NGOs) are not effective enough. The involvement of these agents is essential in the initial stages, but our research showed that peer influence and the caste system have a stronger influence thereafter. Thus, any strategy should consider the involvement of both and the different role they might play in the process.

Additional opportunities derive from the promotion of the benefits of open-pollinated varieties for human and environmental well-being. Not only do these varieties optimise natural biological processes, as they reduce the need for external inputs that negatively impact human and environmental health, but they also indirectly promote products and practices that are more environmentally sound. Hence, they lie at the starting point of the transition to sustainable agriculture.

Overall, if sustainable agricultural technologies are to be disseminated in Nepal, the challenge is to spread these effective sustainable processes and lessons among all farmers (not only small-scale ones). Governmental policies should address the needs of all of them. The government has different instruments in place, such as its extension system and the programmes for seed distribution and farmer training. They are highly appreciated by farmers and might influence new uses and practices, but they should be more reliable and support both the official and non-official organisations involved.

Finally, we emphasise our cautious optimism about the extent of our results, since they derive from the analysis of a single case, even if strategically selected to fulfil the objectives of our research. As we have defended throughout this study, perceptions are context-specific, and as such not transferable. However, the results might be of interest not only for Nepal, but also for other countries addressing similar challenges.

**Author Contributions:** Conceptualization, D.F. and M.M.D.-S.; investigation, D.F.; methodology, D.F. and M.M.D.-S.; supervision, M.M.D.-S.; validation, M.M.D.-S.; writing—original draft, D.F.; writing—review and editing, M.M.D.-S. All authors have read and agreed to the published version of the manuscript.

**Funding:** This research received no external funding.

**Acknowledgments:** This research would not have been possible without an IMRD Consortium Scholarship, funded by the European Commission, which provided financial support for Dana Freshley for the duration of her International Master in Rural Development Programme. We received support from several people on the ground in Nepal, without whose help this research would not have been possible. First of all, we thank Sijal Pokharel for her endless assistance with translations and guidance, both on and off the field; Gauri Shrestha for generously hosting us, for introducing us to farmers and for escorting us to interview locations; finally, Jaya Mukunda Khanal, Gopal Shrestha and Indra Raj Pandey of CEAPRED, who provided formal assistance and endorsement for the duration of the field research.

**Conflicts of Interest:** The authors declare no conflicts of interest.

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
