# Peer review of "Learning from the Past in the Transition to Open-Pollinated Varieties"

_sustainability, doi:10.3390/su12114716_

Round 1

Reviewer 1 Report

The introduction is well elaborated. In defining the goal of the research authors decided to learn on the hybrid seed spread success and to transfer that model on open-pollinated (O-P) varieties.

Authors claimed: „ Our research identified aspects related to social, economic, environmental and institutional factors that might support this shift in Nepal.“, but most of the aspects are described very generally. The paper is mostly based on qualitative analysis and literature review, and, unfortunately, models of open-pollinated varieties introduction are missing. Especially those related strickly to Nepal?

Most of my comments and suggestion are devoted to cost-benefit aspects of the switch from hybrid to local varieties.

In the beginning, it would be beneficially for the readers to learn a more about Nepal agriculture in general and size of the seed market.

Instead of, or in parallel, with the of nationality, religion and literacy rate, the study site is should be described in the terms of utilized agricultural area, production structure…Agricultural statistics in developing countries could be challenging, but authors are welcomed to provide some rough estimates too.

Are there any significant differences in farm size, farm types income, education level… between farmers in the valley and on the hills?

Table 1 is too long and hard to read. What kind of production are we talking about (table 1)? Could you name some of the produce? Table 1, second column „Percentage of crops“ refers to the percentage of the area utilised, percentage of the costs of seeds, the percentage in the total yield, or something else?

I supposed that hybrid seed comes from multinational companies? What about O-P seeds? Could farmers find a seed of O-P varieties at the market? Is the switch from hybrid to local varieties currently even possible and to what extent? Could you provide quantitative analysis in the terms of market potential, amounts of seed sold and similar? What is the price relationship between hybrid and O-P seeds

If "...economic gains were major reasons for hybrid seed adoption" then the paper should present economic aspects of open-pollinated seed and compare it with hybrid seeds. Based on the representative crops for the study site, a cost-benefit analysis should be produced.  It should encompass economic factors like costs and farm income (major yield gains), land price… and environmental benefit and costs into the analysis? Influence on the value chain could be estimated too. When talking marketing aspects could you quantify the following statement: "Another personal challenge for farmers was the loss of taste and nutrition of hybrid seeds. Farmers mentioned that hybrid products profoundly missed the taste and nutritional quality of local varieties. For this reason, even after adopting hybrid seeds for commercial cultivation, many farmers kept a small plot of land to grow local crops for their own consumption. Farmers tended to use hybrid crops to feed their livestock but not to feed themselves."

And, environmental aspects. Is a decrease in soil productivity consequence of soil management practices or type of seed? When talking about environment could we find a connection between the type of seed and biodiversity? 

Technical issues

Line 36-38 In the matter of fact, these are FAO numbers [4]? But, I'm more concerned with the appropriate use of a percentage. I believe that the use of number like 200% is not correct, and would prefer expression like „Yield doubled or yield increased 3.5 times “. Please check this with the editor.

Line 152-157- Does not belong to Methods, but results.

Line 161 What is CEAPRED? Please explain the meaning of the abbreviation when it appears for the first time in the text.

231-232 „He explained that farmers do not understand that open-pollinated varieties have the potential to be profitable in the long term.“ How did he explain it?

Reviewer 2 Report

I have mixed feelings about this article.
How am I supposed to treat him? As a scientific article or as a polygraphic-scientific article?
If I would evaluate it scientifically, my review is negative. First of all, there is no scientific purpose and justification. The authors have not justified why they are writing this article. For me it is a survey report and not a scientific article.
The described method of data collection is also not accurate and I have a feeling that the authors present their results in a subjective and not objective way. The analysis of the obtained results is read as an article in the press and not as a scientific study supported by statistical methods and deep inferences.
Conclusions are also laconic.
However, if it is to be a popular scientific article, the Editorial Board will qualify it anyway, I agree to publish it under certain conditions. The authors must specify the purpose of the research and describe the research method in more detail as well as the results obtained. It is necessary to focus more on the analysis of factors influencing the use of hybrid seeds, to group them precisely, apply some statistical test and refine the conclusions.
The final decision is left to the Editorial Office.

Reviewer 3 Report

The article is within the scope of the journal and is well written. There is one major weakness that I have highlighted, and authors may like to consider it in revising the manuscript in their response.

The method is not clearly explained and hence it is difficult to comment on the results.

There is need to describe various terms ‘castes’, open pollinated, hybrid etc.

Specific comments.

Introduction is well written and referenced. Some parts like line 105 may go into method section along with some other information about the case study site.

Line 135: It will be good to know how many interviews and focus groups etc here. Something on farmer selection will be helpful.

Line 170: It will be good to know more on this process. How coding was done? Did only one author did it or both participated in this data analysis?

Line 187: Diffusion process is not described in method section. Please describe. The main results are summarised in Table 1. It is not clear how authors arrive at these results.

It will be good to describe all factors – internal, external and environmental ones in the method section.

Discussion is well balanced and may improve if there is brief discussion on limitations of diffusion process, low number of respondents and once case study site etc. Are these results representative of the region?

Round 2

Reviewer 1 Report

I would like to thank the authors of the improvements they made.

The paper, in its current form, is a well-presented piece of qualitative research.

But, I still have a certain reserve about the paper. Some information still hasn’t been adequately supported. I’m not insisting on cost-benefit analysis for the representative crop, as suggested before, but requiring stronger support for the text (italic) below within the paper.

Economic benefits from open-pollinated varieties derive from higher income (due to the higher price I suppose? Not yield?) and lower costs?  Is this really a case in Nepal, or the situation is more complex? Reading this paper over again, I can’t tell are high environmental costs of hybrid seeds (I haven’t found any data of that in Nepal?) or dependency of hybrid seed imports, or both, the source of government interest in open-pollinated varieties?

The authors introduced the subchapter "Main limitations of the research". It is praiseworthy and contributes to the quality of the paper. Nevertheless, numeration is wrong. It is not 2.5 (line 576), but 4.5, I suppose?

Authors and editors should discuss table 1 and a few other technical issues mentioned in my first review of the paper. Table 1 could be split into two tables- Valley and Hillside. 

Reviewer 2 Report

Thank you for the materials you sent in.
The authors have convinced me that they use the right research methods. I represent technical sciences, where we look at certain methods differently.
The changes introduced to the article have significantly improved the quality of the article.
Greetings to .

Author Response

Dear reviewer

We really appreciate your positive comments and feel honored you understand our view

Reviewer 3 Report

Thank you for the revisions. 

Author Response

Dear reviewer

Thanks for your support in the previous round that highly contributed to improve our manuscript